# Metabolic Syndrome and Endocrine Disrupting Chemicals: An Overview of Exposure and Health Effects

**DOI:** 10.3390/ijerph182413047

**Published:** 2021-12-10

**Authors:** Elsi Haverinen, Mariana F. Fernandez, Vicente Mustieles, Hanna Tolonen

**Affiliations:** 1Department of Public Health and Welfare, Finnish Institute for Health and Welfare (THL), 00300 Helsinki, Finland; elsi.haverinen@thl.fi; 2Department of Radiology, School of Medicine, University of Granada, 18016 Granada, Spain; marieta@ugr.es (M.F.F.); vmustieles@ugr.es (V.M.); 3Center of Biomedical Research (CIBM), University of Granada, 18016 Granada, Spain; 4Consortium for Biomedical Research and Epidemiology & Public Health (CIBERESP), 28029 Madrid, Spain

**Keywords:** metabolic syndrome, endocrine disrupting chemicals, human biomonitoring, HBM4EU, obesity, insulin resistance, diabetes, dyslipidemia, hypertension

## Abstract

Increasing prevalence of metabolic syndrome (MetS) is causing a significant health burden among the European population. Current knowledge supports the notion that endocrine-disrupting chemicals (EDCs) interfere with human metabolism and hormonal balance, contributing to the conventionally recognized lifestyle-related MetS risk factors. This review aims to identify epidemiological studies focusing on the association between MetS or its individual components (e.g., obesity, insulin resistance, diabetes, dyslipidemia and hypertension) and eight HBM4EU priority substances (bisphenol A (BPA), per- and polyfluoroalkyl substances (PFASs), phthalates, polycyclic aromatic hydrocarbons (PAHs), pesticides and heavy metals (cadmium, arsenic and mercury)). Thus far, human biomonitoring (HBM) studies have presented evidence supporting the role of EDC exposures on the development of individual MetS components. The strength of the association varies between the components and EDCs. Current evidence on metabolic disturbances and EDCs is still limited and heterogeneous, and mainly represent studies from North America and Asia, highlighting the need for well-conducted and harmonized HBM programmes among the European population. Rigorous and ongoing HBM in combination with health monitoring can help to identify the most concerning EDC exposures, to guide future risk assessment and policy actions.

## 1. Introduction

The burden of non-communicable diseases (NCDs) is increasing worldwide and expanding rapidly, affecting not only adults but also children and adolescents [1]. Many NCDs share common risk factors such as sedentary lifestyles and unhealthy diets, increasing the risk of obesity, hypertension, and distorted lipid and glucose metabolism, which together are also known as metabolic syndrome (MetS), a strong predictor of cardiovascular disease morbidity and mortality. 

The latest MetS definition is from the Joint Interim Statement (JIS), developed jointly by the International Diabetes Federation, the National Heart, Lung, and Blood Institute, the American Heart Association, the World Heart Federation, the International Atherosclerosis Society, and the International Association for the Study of Obesity [2]. The JIS definition of MetS requires that at least three of the following five clinical findings are met: elevated waist circumference, elevated triglycerides, reduced HDL cholesterol, elevated blood pressure, and/or elevated fasting glucose [2]. In clinical practice and research, several definitions of MetS with different cut-off points are still used, which may change classifications, analysis, results, and interpretation of the findings [3,4]. 

Traditionally, MetS has been related to unhealthy lifestyle factors, such as high calorie and ultra-processed diets, decreased physical activity, and genetic predisposition. [5]. Since environmental chemical production has increased rapidly in the past years [6], interest has also grown towards understanding the association of chemical exposure and aetiology and pathophysiology of metabolic disturbances through endocrine disruption. Therefore, recently, exposure to endocrine disrupting chemicals (EDCs) has been identified as an additional and inadvertent risk factor for metabolic disorders. EDCs exposure starts in utero and continues throughout the human lifespan. Early life exposure, pregnancy and childhood have been identified as high vulnerability stages for EDC exposure, increasing the risk of disease later in life and in subsequent generations [7]. Other identified high-risk groups are certain occupations [8,9,10], and those with low socioeconomic status [11].

The Endocrine Society has defined EDCs as “exogenous chemicals, or mixture of chemicals, that can interfere with any aspect of hormone action” [12]. The WHO defined endocrine disruptors as: “an exogenous substance or mixture that alters function(s) of the endocrine system and consequently causes adverse health effects in an intact organism, or its progeny, or (sub)-populations” [13]. 

Several European studies have estimated the prevalence of MetS to be between 13–35% among large European cohorts, mostly around 25% [3,14,15]. In addition to the considerable public health burden and increased risk of cardiovascular diseases, type 2 diabetes and non-alcoholic fatty liver disease [5], the economic impact of MetS on Europe’s health care costs is tremendous [16,17]. EDC-related health costs in the European Union have been assessed to be several hundreds of billions of euros annually [17]. Prenatal BPA exposure alone was identified to have a 20–69% probability of causing 42,400 cases of childhood obesity, with associated lifetime costs of EUR 1.54 billion [18].

In this overview, the most suggestive EDCs prioritized in the European Human Biomonitoring Initiative (HBM4EU) [19,20] are presented in relation to the MetS components. The aim of this review is not to be all-encompassing, but rather focus on the most crucial EDCs identified and provide information for policy makers, the general public and health professionals towards a more comprehensive approach regarding metabolic disturbances.

## 2. Methods

The HBM4EU is a joint effort of 30 countries and the European Environment Agency, co-funded by the European Commission. The initiative coordinates and advances HBM in Europe among different populations, identifies possible health impacts caused by environmental chemicals, and provides information on possible health effects to support policy making. During the HBM4EU initiative 2017–2021, two sets of priority substances were identified by international research teams, substance experts and other public health professionals. [19,20] This prioritization led to the identification of 18 compounds or groups of substances: acrylamide, aniline family, aprotic solvents, arsenic, benzophenones, bisphenols, cadmium (Cd), chromium VI (Cr VI), flame retardants, lead (Ld), mercury (Hg), mycotoxins, per-/polyfluorinated compounds (PFAS), non-persistent pesticides, phthalates and Hexamoll^®^DINCH, polycyclic aromatic hydrocarbons (PAHs), chemical mixtures, and emerging chemicals. These substances are presented in scoping documents, which can be found on the project’s website with detailed information [21]. 

Those scoping documents were used as basis for this review. A supplementary literature search on specific HBM4EU priority substances and metabolic syndrome and its components was conducted during the first half of 2021 on PubMed. In the literature search, key words, such as ‘chemical exposure’, ‘environmental chemicals’ and each of the individual priority substances or groups of substances and ‘metabolic syndrome’ and each of its individual components ‘diabetes’, ‘glucose’, ‘obesity’, ‘weight’, ’blood pressure’, ‘hypertension’, ‘lipids’ were used in different combinations. The search criteria were limited to recent systematic reviews, reviews, and meta-analysis. If there were no or a very limited number of review papers available, individual epidemiological studies were also included in the review. Through this process we identified eight substances presented in this study for which associations with MetS or any of its components were observed.

This was not a systematic review, but a scoping review methodology was applied. A scoping review provides an overview of the available research evidence but does not provide a quantitative assessment and does not allow use of formal meta-analytic methods [22].

In this short overview, only the most suggestive associations between the HBM4EU selected chemicals and MetS outcomes are presented. The main focus was on vulnerable populations, such as pregnant women, in utero exposure and childhood exposure.

## 3. Results

Figure 1 summarizes the existing knowledge on associations between different chemical families and MetS components. The following sections will provide more details on existing knowledge of chemicals or groups of substances.

### 3.1. Bisphenols

Bisphenol A (BPA) is one the most produced and used plasticizers worldwide. BPA acts as an endocrine disruptor, xenoestrogen in particular, and the newer substitutes bisphenol F (BPF) and S (BPS) are suggested to have similar effects [23,24]. Human bisphenol exposure is ubiquitous, and contamination happens through ingestion, dermal absorption, and inhalation of particulates and vapor phases [25]. Common exposure pathways include epoxy resins in canned foods/beverages, polycarbonate plastics, thermal paper, plastic toys, dental materials, and consumer goods [26]. As the use of BPA is decreasing, substitutes such as BPS and BPF are becoming more widely used [24]. However, the current evidence shows that most alternative bisphenols are as hormonally active as BPA [24,27].

BPA presents a short biological half-life in the human body, approximately 6 h, and after oral administration the maximum concentration in blood occurs within 80 min [28]. Although rapidly excreted in urine, population BPA exposure is continuous due to its use in myriads of everyday consumer products. To protect against BPA exposure, the European Commission has taken actions by banning the use of BPA in infant feeding bottles [29] and restricting the use of BPA in certain food-contact materials [30]. The amount of BPA allowed in children’s toys has also been further limited down to 0.04 mg/L in toys [21,31]. Additionally, since 2016, the use of BPA has been restricted in thermal paper and the ban took effect in 2020 within the European Union [32]. Although the relevance of BPA exposure and its effects on human health have been widely agreed on, there is a lack of consensus on the cut-off values (tolerable daily intake limits) by different agencies [21].

Several systematic reviews have been conducted to evaluate the current knowledge on BPA exposure and metabolic and glucose disturbances, and they have revealed strong evidence on BPA’s association with metabolic syndrome [28,33,34,35,36]. Most often the research has focused on obesity and glucose disturbances. BPA exposure has been associated with development of hypertension in several studies [37,38,39]. For BPS and BPF, less research thus far exists, however some evidence of BPS’s effect on hypertension has been observed [38]. Teppala et al. [40] investigated the cross-sectional association of BPA exposure and MetS among 2104 NHANES participants during 2003–2008. In the analysis, BPA was positively associated with MetS risk, and the observed association was independent of confounding factors (age, gender, ethnicity, smoking, alcohol intake, physical activity levels and urinary creatinine). 

Recent meta-analyses support that an association exist between BPA exposure and obesity in both children and adults [41,42]. Although limited, preliminary evidence with bisphenol substitutes points to the same direction. BPF has been shown to have some evidence of association between children’s and adolescent’s obesity, interestingly, showing a stronger association among boys, which might suggest possible sex differences [27]. In a study based on the 2013-2016 NHANES survey, BPA was not significantly associated with obesity, in contrast to BPS and BPF [16]. 

Although BPA has been associated with childhood obesity in a previous meta-analysis [41], associations with markers of lipid and glucose homeostasis have been more inconsistent [43]. Notwithstanding, more consistent BPA associations with leptin and adiponectin suggest that adipokines may be more sensitive biomarkers of early metabolic impairment among children [43].

Pregnancy and perinatal periods have been proposed to be particularly susceptible for BPA action and development of type 2 diabetes (T2DM) [44]. Among pregnant women, exposure to BPA was associated with higher systolic and diastolic blood pressure [45]. Urinary BPA concentrations in the second trimester, but not the first trimester of pregnancy, were positively associated with blood glucose levels 1 h after a 50 g glucose tolerance test at 24–28 weeks of gestation among sub-fertile women [46]. Other studies have not found associations with gestational diabetes [47]. In a study investigating maternal bisphenol concentrations and children’s lipid metabolism, no association was discovered with non-fasting lipid concentrations during childhood [48].

In adults, the relationship between BPA and glucose homeostasis has been relatively well-studied. In general, BPA has been associated with increased glycated hemoglobin (HbA1c) levels in adults [49,50], greater serum insulin levels and insulin resistance [51], and higher risk of prediabetes [52] and type 2 diabetes [53]. Moreover, a prospective study identified a susceptible group of adults for BPA effects on glucose homeostasis based on a genetic risk score [54].

Although not without some inconsistencies, the overall toxicological, observational, and human intervention streams of evidence support that BPA exposure during development, but also at adulthood, constitutes a risk factor for obesity, insulin resistance, and other MetS components. Preliminary evidence suggests that BPS and BPF do not constitute safe alternatives. 

### 3.2. Per- and Polyfluoroalkyl Substances

Per- and polyfluoroalkyl substances (PFASs) are man-made chemicals that have been produced globally since the 1950s. PFAS are a group of synthetic fluorinated compounds, which are widely used in industrial and consumer products including stain- and water-resistant coatings for fabrics and carpets, oil-resistant coatings for paper products approved for food contact, floor polishes, pesticide formulations, fire-fighting foams, and mining and oil well surfactants. The European Food and Safety Authority (EFSA) CONTAM Panel has set up the tolerable weekly intake (TWI) for perfluorooctanoic acid (PFOA), perfluorononanoic acid (PFNA), perfluorohexanesulfonic acid (PFHxS), and perfluorooctanesulfonic acid (PFOS) to be 4.4 ng/kg body weight (bw) per week [55].

The most common and well-researched forms of PFAS are PFOA and PFOS. PFOA and PFOS are long-chain perfluorinated compounds, which have shown carcinogenic, reprotoxic, and immunotoxic features, as well as being capable of affecting thyroid and lipid metabolism [56,57,58]. Several other long-chain PFAS have also been identified as being highly persistent, bio-accumulative and toxic, as their elimination times go from years to decades in humans [55]. Moreover, there are also concerns about short-chain PFASs, which have been assumed to be less bio-accumulative, however are still persistent and found in drinking water and food, including vegetables [21].

In a recent review [59] focusing on epidemiologic evidence on associations between exposure to PFAS and the development of obesity, diabetes, and non-alcoholic fatty liver disease, some evidence of association was discovered. From the total of 55 studies, approximately 2/3 reported positive associations between PFASs and the prevalence of obesity and/or type 1, type 2, or gestational diabetes [59]. For children, the evidence seems to be less conclusive, as only development of dyslipidemia suggested positive association with PFAS exposure [60]. 

MetS as an outcome was examined in one study, revealing that PFNA was associated with increased risk of MetS, increased waist circumference, elevated triglyceride, and decreased HDL when controlling for multiple PFASs. Although perfluorodecanoic acid (PFDA), PFOA, 2-(N-methyl-PFOSA) acetate (MPAH), and perfluoroundecanoic acid (PFUnDA) were associated with decreased risk of certain MetS components, the most consistent pattern was shown by PFUnDA. [61] Among overweight and obese participants, PFASs have been associated with higher apoB and apoC-III concentrations, but not with total cholesterol or triglycerides [58].

When examining vulnerable populations, children have shown higher serum concentrations of PFAS compared to adults [62]. In utero exposure for PFOA has been linked to children’s obesity [63,64]. Among adolescents, PFOS, PFNA, PFDA, and PFUnDA serum concentrations were positively associated with apolipoprotein B and total and LDL cholesterol [65]. PFAS, PFHxS, PFOS, and PFOA concentrations were positively associated with the risk of hypertension and, furthermore, PFHxS and perfluoroheptane sulfonic acid (PFHpS) concentrations were positively associated with obesity [65]. No association was discovered between PFAS exposure and development of hypertension in children [66].

Among pregnant women, gestational diabetes mellitus (GDM) and PFAS exposure has so far revealed inconsistent and component specific differences [67,68,69]. In one study, PFOS, PFOA, PFHxS, PFNA, 2-(N-ethyl-perfluorooctane sulfonamide) acetate, N-ethyl perfluorooctanesulfonamidoacetate (EtFOSAA), N-methyl perfluorooctanesulfon-aminodoacetate (MeFOSAA), perfluorodecanoate, and perfluorooctanesulfonamidoacetate (FOSAA) exposure showed no association with glucose tolerance and PFAS exposure [68]. Two other studies showed that the serum levels of perfluorobutane sulfonate (PFBS) and perfluorododecanoid acid (PFDoA) were significantly higher in the GDM group in comparison to the controls [69], and in another study PFOS and PFHxS were associated with impaired glucose intolerance or gestational diabetes mellitus [67]. A possible association between pregnancy-induced hypertension (PIH) and PFAS exposure was discovered in one study [70], as another study revealed high detection rate of PFASs in the placenta but no evidence on hypertensive disorders during pregnancy [71]. PFOA has been further associated with increased maternal total cholesterol [67].

As PFASs have shown increased association with diverse adverse health outcomes, new substitutes are constantly entering the market. For example, 2,3,3,3-Tetra-fluoro-2-(heptafluoropropoxy) propavoic acid (GenX) has been developed to substitute PFOA use. GenX has shown some evidence of metabolic disturbances, however to a lesser extent than PFASs [72]. Another PFAS substitute, chlorinated polyfluorinated ether sulfonic acids (Cl-PFESA), was associated with increased prevalence of hypertension and elevated DBP. Moreover, it seemed that women were more susceptible to changes associated with Cl-PFESA [73]. However, knowledge regarding exposure to these compounds and adverse health effects is still limited.

Recently, the state of knowledge on the effect of PFAS (mainly PFOS and PFOA) exposure on cholesterol and triglyceride homeostasis has been reviewed [74]. Many epidemiological studies show positive associations between increased blood levels of total blood cholesterol, and in some cases triglyceride with higher PFOS/PFOA levels, but most of these are cross-sectional studies. In vitro research in human liver cells shows that PFOS/PFOA activate the PPARα pathway, as well as some other nuclear receptors, such as PXR. In addition, data indicate that suppression of the nuclear receptor HNF4α signaling pathway as well as disturbances of bile acid metabolism and transport could be important molecular events that require, however, further investigation. Experimental studies (mainly rodents) exposed to high levels of PFAS (at least 100-times higher than in humans), however, show reverse effects.

### 3.3. Phthalates

Phthalates are the most used plasticizers worldwide, with a consumption of 7.5 million tons annually. They are included in numerous products, including vinyl flooring, adhesives, detergents, lubricating oils, automotive plastics, children’s toys, textiles, wallpapers, food packaging, and personal care products. Phthalates are known to exhibit a variety of health effects. Notably, not all phthalates have the same endocrine-disrupting potency or developmental effects, nor are effects of exposure to their mixtures fully understood [21,75]. 

In HBM studies, the phthalate metabolites examined vary between the studies. Due to their adverse health effects on reproduction and development seen in animal studies, which can also be considered relevant to humans, the European Union has restricted the amount of diiso-decyl phthalate (DiDP), di-n-octyl phthalate (DnOP) and di-iso-nonyl phthalate (DiNP) in children’s’ toys and items that can be placed in children’s mouths [76]. The use of s di(2-ethylhexyl) phthalate (DEHP), butylbenzyl phthalate (BBzP), di-n-butyl phthalate (DnBP), and diiso-butyl phthalate (DiBP) are additionally under restriction [77] and must not be used without permission. In addition, DEHP, DnBP, BBzP, DiBP, bis(2-Methoxyethyl) phthalate (DMEP), di-n/iso-pentyl phthalate (DnPeP, DiPeP), and 1,2-benzenedicarboxylic acid (DHNUP) are further prohibited for use in cosmetics in the European Union [78]. Additionally, new chemical compositions with lower toxicity have been developed to be used as alternatives for the traditional phthalates, e.g., Hexamoll® DINCH® [21]. In the HBM4EU, HBM guidance values (HBM-GVs) were established for specific phthalates (DEHP, BBzP, DnBP, DiBP, di(2-propylheptyl) phthalate (DPHP) and DINCH) and for children, adults including adolescents, and occupational exposure population groups [79,80]. 

In general, the available review literature supports a positive association between phthalates and obesity-related factors [28,81,82], glucose disturbances [28,33,82,83], and hypertension [84]. A review on phthalates and gestational metabolic syndrome (GMS) reported inconclusive results [85]. Among children, a systematic review’s meta-analysis indicated a significant association between individual phthalate metabolites with body mass index (BMI), BMI z-score, waist circumference, dyslipidemia, and glucose in serum. In addition, significant associations were observed between prenatal exposure to some phthalate metabolites and birth weight [81].

Possible sex differences in phthalate exposure and development of MetS have been observed [86,87,88,89,90]. For men, higher DEHP metabolite concentrations were associated with increased odds of MetS, while in women, the highest association was observed with monobenzyl phthalate (MBzP) the main metabolite of BBzP [88]. Additionally, increases in waist circumference and BMI have been linked to DEHP, BBzP, DnBP and diethyl phthalate (DEP) exposure in men [90] and DEP exposure in female adolescents and adults [87]. Among pregnant women, decreased systolic and/or diastolic blood pressure has been associated with higher phthalate levels [45]. 

Among children, some associations between phthalate exposure and development of hypertension have been reported [91,92]. For lipids, one study examining children and adolescents observed no association with phthalate metabolites and triglycerides or high-density lipoproteins [91]. Another study observed lower levels of total and low-density lipoprotein (LDL-C) in response to higher urinary phthalate metabolites (DEP and DnBP metabolites and MCPP, a metabolite for various phthalate compounds) in boys, and lower LDL-C and di-2-ethylhexyl phthalate (ΣDEHP) in girls [89]. Adolescents with intermediate monobutyl phthalate (MnBP) concentrations also presented higher odds of MetS in comparison to adolescents with lower concentrations; interestingly, monoisobutyl phthalate (MiBP) concentrations and odds of MetS varied by sex [86]. Furthermore, phthalate exposure during childhood has been associated with lower systolic and diastolic blood pressure [45].

### 3.4. Polycyclic Aromatic Hydrocarbons (PAH)

Polycyclic aromatic hydrocarbons (PAHs) are a group of chemicals composed of carbon and hydrogen atoms, and they are omnipresent environmental pollutants found in the air, water and soil. PAHs are a result of the incomplete burning of coal, oil and gas, in car emissions and tobacco smoke. PAHs are spread through the atmosphere, especially in close proximity to roads with heavy traffic, municipal waste incinerators, and different industrial sites. In addition to air exposure, PAHs can be found on household items, such as cosmetics, coatings, and rubber. [93] Additionally, PAHs can also be generated and ingested through foods cooked at high temperatures. 

Main exposure route in humans is inhalation but also some exposure through ingestion and dermal absorption occurs. Most vulnerable populations to PAH exposure are smokers, certain industry workers and children [94,95]. Most often, HBM studies measure PAH metabolites in urine samples [21].

PAHs are regulated based on the National Emission Ceilings Directive 2001/81/EC [96]. The HBM4EU initiative provides a comprehensive presentation of the legislative framework regarding PAHs and other air pollutants [21].

One systematic review and meta-analysis was identified examining associations between urinary PAH metabolites and development of diabetes, revealing significantly higher pooled odds of T2DM in the highest group in comparison to the lowest category of urinary naphthalene, fluorine, phenanthrene, and total polycyclic aromatic hydrocarbon (OH-PAH) metabolites [97]. Another systematic review observed positive associations between PAH exposure with increased risk of elevated blood pressure and obesity [98]. The associations between hypertension and PAH exposure have been observed in several previous observational studies [99,100,101,102,103,104,105,106,107,108,109,110,111,112,113]. In two NHANES studies, research on metabolic disturbances and PAH exposure show evidence of the association between obesity (2-hydroxynaphthalene (2-NAP), BMI and 2-NAP, 2-hydrozyfluorene (2-FLUO), 3-hydrozyfluorine (3-FLUO) and 2-hydroxyphenanthrene (2-PHEN)), type 2 diabetes (1-hydroxynaphthalene (1-NAP), 2-NAP, 2-PHEN and 1-pyrene), dyslipidemia (1-NAP, 2-NAP, 2-FLUO, 3-FLUO and 2-PHEN), and hypertension (2-NAP and 2-PHEN) [101,104]. 

Occupational exposure to PAHs has been observed among certain occupations and industries. Coke oven workers were shown to have higher urinary levels of 1-NAP and 2-FLUO in a significant dose–response relationship, with increased prevalence of MetS and, furthermore, 1-NAP was positively associated with low HDL-C [105]. Elevated urinary 4-hydroxyphenanthrene (4-OHPh) was significantly associated with increased risk of T2DM [106]. Among outdoor workers, hydroxypyrene (1-HOP) occupational exposure was negatively associated with both systolic and diastolic blood pressure [107].

Among children, exposures to 2-naphtol, 9-phenanthrol, and total OH-PAH have been associated with increased risk of obesity, and exposure to 1-HOP has been related with higher risk of cardiometabolic risk factors in children who had excess weight [108]. Another study observed that BMI z-score, waist circumference, and obesity were positively associated with total PAH and napthalene metabolites in children. [109]. In addition, boys attending a school close to an oil refinery showed an increased the risk of prehypertension [110].

### 3.5. Pesticides

Pesticide use is ubiquitous worldwide, however evidence on endocrine-disrupting human health effects is still scarce. The synthetic pesticides are classified in four main groups: organochlorines, organophosphorus, carbamates and pyrethroids [111]. Currently, pyrethroids are the most used pesticides in the EU. Studies of pyrethroids are scarce, and therefore a complete picture of low-level environmental and dietary exposure effects on human health is difficult to achieve. [21,111].

Humans are exposed to pesticides via ingestion, inhalation, or dermal absorption through skin. For the general population, pesticide residues in food constitute the main source of exposure. The most vulnerable populations include infants, children, pregnant women, agriculture farm workers, and pesticide applicators. [21,111] Urine, blood/serum, and hair are used as biomonitoring matrices. [21]

An EU Pesticides Database has been developed to provide information on active substances used in plant protection products, including Maximum Residue Levels (MRLs) in food products, and emergency authorizations of plant protection products in the Member States (https://ec.europa.eu/food/plants/pesticides/eu-pesticides-database_en). As children and pregnant women are considered the most vulnerable populations for exposure, baby food products are included in the control programs and the European Commission has defined specific rules for foods specially manufactured for infants (below 12 months of age) and young children (between 1 and 3 years of age) [112].

Two systematic reviews (one including a meta-analysis) and one review article showed some evidence of non-persistent pesticide exposure and diabetes or other glucose disturbances (insulin resistance, beta-cell dysfunction) [113,114,115]. Both systematic reviews identified current limitations of the evidence: high heterogeneity, cross-sectional study designs, lack of addressing selection bias, addressing confounding factors and wide confidence intervals. In addition, to glucose disturbances, Xiao et al. [115] identified four obesity-related studies, indicating inconsistent results.

MetS as an individual health outcome has been examined in three studies [116,117,118]. In one cross-sectional study [118], the NCEP-ATP III definition was used as criteria for MetS. Seven pesticides showed significant associations with MetS across the quintiles of exposure in each of the unadjusted models (p,p′-DDT, p,p′-DDE, HCB, β-HCCH, oxychlor, tNONA, Mirex). In a case-control study [117] it was found that after adjusting for all other confounders except for BMI, beta-hexachlorocyclohexane (β-HCH) and heptachlor epoxide were positively associated with MetS, and in a cohort study [116] participants who had been exposed to pesticides had an elevated incidence of MetS in comparison to the non-exposed. To date, this has been the only study to assess the relationship between incidence of MetS and pesticide exposure. [116] Organochlorine pesticides are often measured in blood, although adipose tissue is known to contain a reservoir of accumulated concentrations that may better capture long-term exposure [36]. It has been shown that β-HCH and hexachlorobenzene (HCB) were associated with the risk of suffering at least one MetS risk factor [119].

In two cohort studies [120,121], inconclusive results were obtained on organochlorine pesticides (OCPs), as some OC pesticides and PCBs predicted excess adiposity, dyslipidemia, and insulin resistance among participants without diabetes [120], while no association was detected in lipid outcomes in the other study [121]. Although associations have not always been consistent, it has been recognized that organochlorine pesticides constitute a risk factor for metabolic disturbances including excess adiposity, dyslipidemia, and insulin resistance in adults and possibly in children in response to prenatal exposure [122,123,124].

Limited data on pesticide exposure and pregnant women’s hypertension exist, as only suggestive findings exist thus far [125,126]. Additionally, a very limited number of studies have so far been conducted on human early life exposure and pesticides, however animal studies indicate some evidence of metabolic disturbances [127]. A prenatal exposure study discovered that prenatal dichlorodiphenyltrichloroethane (DDT) exposure and its metabolite dichlorodiphenyldichloroethylene (DDE) were associated with hypertension over a long follow-up period [128].

### 3.6. Heavy Metals

#### 3.6.1. Arsenic (As)

Arsenic is a natural component on earth’s crust, as it can be found in soil and groundwater in number of countries. The toxicity of arsenic depends on its form (inorganic or organic) and species. [129] The toxic species are inorganic forms, such as arsenious acid (As[III]), arsenic acid (As[V]), monomethylarsonic acid (MMA), dimethylarsinic acid (DMA), and trimethylarsine oxide (TMAO). On the other hand, organic arsenic, which is consumed in food (mainly fish and seafood) is less harmful for human health. The sources for exposure are both natural and anthropogenic [21]. Usually, exposure happens through contaminated drinking water, by using water in food preparation, through irrigation of crops, in different industrial processes, by eating contaminated food, or smoking tobacco [130]. Furthermore, occupational exposure occurs in industries, including gold mining, arsenic production, wood preservation, glass manufacturing, and smelting operations [131,132,133,134]. 

Arsenic has been shown to have severe health effects from short- and long-term exposure. Arsenic exposure in children is similar to adults, and they do not seem to be more sensitive [130].

Arsenic exposure is mostly measured in spot urine samples that can detect inorganic arsenic (iAs) and methylated metabolites (DMA and MMA). Urine is considered as a preferred collection method for exposure, due to its non-invasive nature, easiness of a collection, and good detection of metabolites. Blood sampling is also possible, however is less used due to its invasiveness and the short half-life of inorganic and organic arsenic species. The safe total daily intake (TDI) limit ranges between 20-300 micrograms/per day [31].

Development of obesity and arsenic exposure showed inconsistent results in two review articles due to differences in dose, form, and route of exposure [135,136]. For diabetes-related outcomes, some evidence of association has been presented, but notably still a limited number of epidemiological studies exist [83,137]. In a meta-analysis, both low and high exposure levels of arsenic were suggested to have a positive association between exposure and development of hypertension [138]. According to a systematic review, arsenic exposure can affect lipid metabolism by reducing serum HDL levels and increasing serum LDL levels [139]. 

Maternal arsenic exposure and gestational diabetes mellitus (GDM) show indications of association [140,141]. In utero arsenic exposure and development of diabetes has thus far been studied in very limited way, however some evidence of association has been indicated [142]. Among children, development of obesity and other metabolic disturbances (Homeostatic Model Assessment for Insulin Resistance (HOMA-IR)) have been linked to children with higher arsenic concentrations [143].

#### 3.6.2. Mercury (Hg)

Mercury is a naturally occurring element that is found in air, water, and soil. Hg exposure can cause several different adverse health effects to humans and is considered as one of the top ten chemicals (or groups of chemicals) of major public health concern. The Minamata Convention on Mercury launched in 2017 is an international commitment in addressing mercury pollution, and it has been ratified by the European Union [144]. In Europe, strict regulations also exist in restricting Hg pollution and human exposure. The European Commission introduced the Community Strategy Concerning Mercury in 2005, which consists of a comprehensive plan to address Hg use and pollution. European legislation concerning Hg includes instructions on food safety, chemicals, environment, consumer products, and occupational health and safety. Two especially vulnerable populations for Hg exposure are fetuses and individuals with occupational exposure [21]. 

Hair, urine, blood, nails, breast milk, cord tissues, cord blood, and placenta are the most used measurement matrixes for mercury exposure. The choice of matrix depends on the time of sampling after exposure, whether chronic or acute exposure is being investigated, and which types of mercury compounds are being assessed [21,145].

Three systematic reviews have shown evidence of metabolic disturbances in response to Hg exposure [146,147,148]. Tinkov et al. [148] discovered six studies on obesity, from which five showed positive association with exposure to mercury. For glucose metabolism/DM, the results were more inconclusive; six studies showed a positive link and three (including one systematic review) found no association in human studies. Additionally, dyslipidemia and atherosclerosis showed inconsistent results. Similar results were discovered in Roy et al.’s [147] systematic review focusing on diabetes, MetS and insulin resistance, and Hg exposure. The authors included 34 studies in their analysis and concluded that there is evidence that suggests a possible association between Hg and incidence of MetS and/or DM. The third systematic review concluded that data from model organisms suggest a possible association between Hg exposure and development of MetS, albeit human data are still inconclusive [146]. Hypertension was examined in a systematic review and meta-analysis, revealing that people with higher concentrations rates of Hg also had higher risk of hypertension in the dose-response analysis [149].

Children’s chronic Hg exposure was assessed in a systematic review from which four of the included articles assessed prenatal exposure, two studied both prenatal and postnatal exposures, and two investigated postnatal exposures. A positive significant association was detected between chronic Hg exposure and increased levels of blood pressure in children or adolescents according to four studies (three of them analyzing prenatal exposure). However, designs of the studies were heterogeneous and different covariates were used in the studies [150]. An individual cohort study suggested that young adults (20–32 years) with high Hg exposure during their young adulthood might have increased risk of DM later in life [151].

#### 3.6.3. Cadmium (Cd)

Cd can be found globally on natural and anthropogenic sources. Most common anthropogenic sources are color pigments and stabilizers used in plastics, automobile radiators, alkaline batteries, mining activities, fertilizers, sewage sludge, and inappropriate waste disposal. Exposure of Cd among adults in Europe happens through food, water, and tobacco smoke, containing around 10-20 micrograms of cadmium per day. Cd-rich foods include seafood, liver, kidney, wild mushrooms, flaxseed, and cocoa powder. However, 80% of the contamination from food comes from cereals, potatoes, and vegetables grown in contaminated soil. High Cd levels in soil are usually due to use of phosphate fertilizers, and in non-polluted areas the concentration levels are relatively low [21].

Among vulnerable populations, such as pregnant women and occupationally exposure adults, some evidence on the adverse effect of Cd on metabolic health exists. The association of increased risk of gestational diabetes (GDM) and Cd seems to be suggestive [152,153,154]. Koreans working with heavy metals were shown to possibly have an increased risk of MetS [155].

Cd exposure and metabolic disturbances have been examined in several reviews, systematic reviews, and meta-analysis. Studies on overweight, obesity, and Cd exposure show contradictory results [156]. For glucose disturbances and T2DM, suggestive associations have been observed [156,157]. Hypertension was examined in three review studies (one systematic review, one systematic review and meta-analysis, and one review article) showing some evidence of association with cadmium exposure [157,158,159]. Satarug et al. [157] highlight that women seemed to be more susceptible to adverse blood pressure effects than men. No systematic reviews were found on lipids, however two individual studies showed evidence of dyslipidemia in accordance with cadmium exposure [160,161].

## 4. Discussion

MetS is a major public health challenge in Europe due to its high prevalence and costs [3,14,15,16,17,18]. Apart from the traditional risk factors such as sedentary lifestyle or unhealthy diet, environmental contaminants are increasingly recognized and studied as additional risk factors [162]. 

As this scoping review highlights, the current evidence on environmental chemicals’ effects on the development of MetS varies between the MetS components and environmental substances. What is inferred from the presented results is that MetS is influenced by the impact of different chemical families on obesity, glucose abnormalities, and blood pressure. Less evidence exists for lipid abnormalities. Among children, impacts on obesity and blood pressure were most common, and more inconsistent results were observed for glucose metabolism than among adults. The results from this review also highlight the effect of chemical mixtures on metabolic outcomes. Although not covered, the different chemical families investigated may act through both similar and dissimilar modes of action, which opens the possibility of additive or even synergic effects [163]. In the context of ubiquitous population exposure to most of the revised chemicals, the concept of mixtures should be kept in mind in both risk assessment and policy making [164]. 

Although not without some inconsistencies, the overall toxicological, observational, and human intervention mainstreams of evidence support that BPA exposure during development, but also at adulthood, constitutes a risk factor for obesity, insulin resistance, and other MetS components. Preliminary evidence suggests that BPS and BPF do not constitute safe alternatives. For PFASs, epidemiological evidence shows an association between obesity, dyslipidemia, and diabetes among adults. In utero exposure has been linked with childhood obesity, and exposure among adolescents with increased risk of obesity, dyslipidemia, and hypertension. Recent evidence on PFAS substitutes GenX and GI-PFESA indicate that they are not safer alternatives in respect to MetS components. For phthalates, reported studies suggest associations with obesity, glucose disturbance, and hypertension among adults. Prenatal exposure is shown to increase risk of childhood obesity, and exposure during the childhood may result increased risk of obesity, lipid, and glucose disturbances. For phthalates, sex differences in the associations have also been observed. Exposure to PAHs has been associated with increased risk of obesity, diabetes, and hypertension. Among children, exposure to PAHs has been linked to increased risk of obesity. For pesticides, existing evidence is still scarce but suggests associations with diabetes and non-persistent pesticides. Furthermore, some evidence to support possible association with MetS has been reported. For three heavy metals (arsenic, mercury, and cadmium), limited evidence exists but there are some indications that exposure would increase the risk of obesity, diabetes, and hypertension. Additionally, for arsenic and mercury, association with the risk of dyslipidemia has been observed. Although several systematic reviews and meta-analysis have been found, most of the included studies are cross-sectional or case-control designs, limiting causal inference. The HBM is an important tool that can help to overcome these limitations. However, at present, most studies are conducted in North America (NHANES) and Asia, and only few a European countries, such as Germany [165] and France [166], have incorporated human biomonitoring modules into their national health examination surveys.

In this review, MetS was examined as an outcome only in few studies, and most available data focused on individual MetS components. Although some MetS components have been widely researched (e.g., obesity and glucose metabolism), others have gained less focus (dyslipidemia and hypertension). Nevertheless, an increase in the risk of isolated MetS components will lead, over time, to a higher risk of MetS and cardiovascular disease, morbidity, and mortality. Therefore, the role of environmental chemicals on MetS should not be overlooked. Indeed, additional studies are needed and should be focused on increasing study quality: harmonization of MetS definitions, harmonization of exposure assessment with certified laboratories undergoing interlaboratory comparisons, representative population samples, longitudinal designs with repeated measurements, harmonization of effect biomarkers implemented and quality control measures, and harmonization of statistical protocols including a selection of the most relevant covariates based on causal graphs [43,167]. All of this will contribute to generating more confident information and improve the uptake of HBM results by risk assessment and policymakers [168].

## 5. Conclusions

Current evidence on the associations of metabolic disturbances and EDC exposures is inconclusive and fragmented, although the overall picture supports the involvement of exposure to many chemical families in the risk of suffering MetS components. There is a need, however, to establish harmonized and standardized HBM procedures among the European population, in addition to rigorous and continuous human biomonitoring combined with health monitoring, including novel effect biomarkers which could provide comprehensive information on EDC exposure and association of metabolic disturbances.

## Figures and Tables

**Figure 1 ijerph-18-13047-f001:**
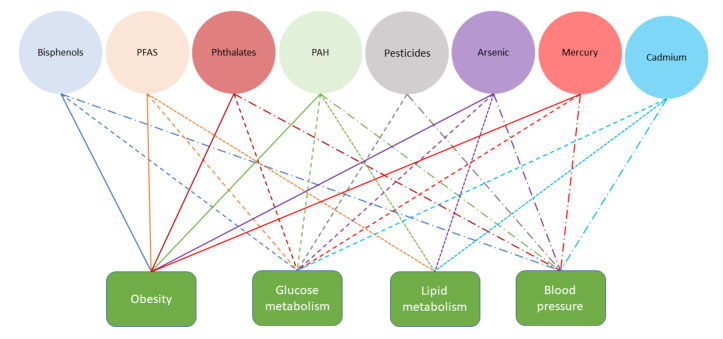
Identified associations between substances and components of metabolic syndrome (color of the line refers to the substance and line type to the component of the metabolic syndrome).

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
