# Peer review of "Metabolic Syndrome and Endocrine Disrupting Chemicals: An Overview of Exposure and Health Effects"

_ijerph, 2021, doi:10.3390/ijerph182413047_

Round 1

Reviewer 1 Report

In this review, the authors mainly talked about the main chemicals potentially causing endocrine alterations (so called endocrine disrupting chemicals) and their relationship with human metabolism, especially the metabolic syndrome and its specific components (i.e.: hypertension, dyslipidemia, obesity, disglycemia). They concluded that the exposure, during different phases of life, to the chemical families investigated, could be a risk factor for suffering of metabolic syndrome or of one or more of its components. Moreover, they argued that there is need to implement well-conducted and harmonized human biomonitoring studies and programs, in order to deeply understand the linkage between these chemicals and endocrine disruptions, thus guiding social and policy actions.

The expression was generally clear, and the content of the review well explained. The article is sufficiently novel and interesting to warrant publication. I recommend minor revisions, mostly concerning grammar mistakes:

  • Line 32: “sendentary” correct in “sedentary
  • Line 51: “EDC” correct in “EDCs
  • Line 66: “[15, 16].” correct in “[15, 16]” (without dot)
  • Line 83: “combounds” correct in “compounds
  • Line 200: “increased WC”. The authors should explain the acronym.
  • Line 309: “temparatures” correct in “temperatures
  • Line 359: “authorisations” correct in “authorizations
  • Line 511: “incluenced” correct in “influenced”
  • Line 534: there are two commas between “definitions” and “harmonization”

Author Response

Thank you for taking time to review our manuscript and providing valuable feedback on it. Please, see below detailed responses to your comments/recommendations.

In this review, the authors mainly talked about the main chemicals potentially causing endocrine alterations (so called endocrine disrupting chemicals) and their relationship with human metabolism, especially the metabolic syndrome and its specific components (i.e.: hypertension, dyslipidemia, obesity, disglycemia). They concluded that the exposure, during different phases of life, to the chemical families investigated, could be a risk factor for suffering of metabolic syndrome or of one or more of its components. Moreover, they argued that there is need to implement well-conducted and harmonized human biomonitoring studies and programs, in order to deeply understand the linkage between these chemicals and endocrine disruptions, thus guiding social and policy actions.

The expression was generally clear, and the content of the review well explained. The article is sufficiently novel and interesting to warrant publication. I recommend minor revisions, mostly concerning grammar mistakes:

  • Line 32: “sendentary” correct in “sedentary

Correction has been done.

  • Line 51: “EDC” correct in “EDCs

Correction has been done.

  • Line 66: “[15, 16].” correct in “[15, 16]” (without dot)

Correction has been done.

  • Line 83: “combounds” correct in “compounds

Correction has been done.

  • Line 200: “increased WC”. The authors should explain the acronym.

Since this the only place where acronym is used, we have now written WC open ‘waist circumference’ and removed unnecessary acronym

  • Line 309: “temparatures” correct in “temperatures

Correction has been done.

  • Line 359: “authorisations” correct in “authorizations

Correction has been done.

  • Line 511: “incluenced” correct in “influenced”

Correction has been done.

  • Line 534: there are two commas between “definitions” and “harmonization”

Correction has been done.

Reviewer 2 Report

Thank you for the review paper, it is a valuable paper which pulls together outcomes specifically relevant to metabolic syndrome and EDC.

My comments on the paper are as follows:

  • First sentence of Introduction (lines 30 and 31), this needs a comma after rapidly. Also please include a reference for this sentence/statement.
  • Line 68 presents a value of 42.400 - this is too accurate. It should be 42.4 at best.
  • Figure 1 - do the different lines (solid and various different types of dashed lines) mean anything?
  • Line 114 - in relation to bisphenols, please clarify if the inhalation pathway relates to vapours or particulates (or both)
  • Section 2 (Methods). This needs a better explanation on how the papers were reviewed. In particular it is noted that the review has summarised outcomes from systematic reviews relevant to the various endpoints relevant to MetS. Have the systematic reviews been done using similar approaches in the different papers? This is important for understanding how comparable the outcomes may be. In addition what is the strength of evidence found in the systematic reviews and can these be supported by biological mechanisms? This is relevant for the review as associations between exposure and an effect can be very different, and the strength of that association and whether that association can be further supported is relevant to being able to translate the outcomes of the systematic review into public health risks.
  • Lines 169 to 173 provides an overview and risk outcomes based on the evidence presented in the section. This is the only chemical where overview/risk outcomes are presented. Could a similar overview discussion (few sentences) be included for all the other chemical sections - this would also support the discussion and conclusions.
  • Line 313, do you mean PAH metabolites in urine samples?
  • Section 3.6.1 should be consistent with the use of the abbreviation "As" for arsenic. It looks a bit odd at the start of a sentence, but some sentences include both the abbreviation and the word, which also looks a bit strange.
  • Is it possible to include a summary table that outlines the strength of the associations identified for each chemical and health outcome - perhaps that is what Figure 1 may have been trying to do, but it would be good to have the outcomes summarised further in the discussion.

Author Response

Thank you for taking time to review our manuscript and providing valuable feedback on it. Please, see below detailed responses to your comments/recommendations.

Thank you for the review paper, it is a valuable paper which pulls together outcomes specifically relevant to metabolic syndrome and EDC.

My comments on the paper are as follows:

  • First sentence of Introduction (lines 30 and 31), this needs a comma after rapidly. Also please include a reference for this sentence/statement.

Comma and reference has been added.

  • Line 68 presents a value of 42.400 - this is too accurate. It should be 42.4 at best.

Here 42.400 is not a proportion but the number of actual cases. 42.400 has been edited to be 42,400.

  • Figure 1 - do the different lines (solid and various different types of dashed lines) mean anything?

Explanation of the colours and line types has been added to the figure.

  • Line 114 - in relation to bisphenols, please clarify if the inhalation pathway relates to vapours or particulates (or both)

Explanation has been added to the text together with the reference.

  • Section 2 (Methods). This needs a better explanation on how the papers were reviewed. In particular it is noted that the review has summarised outcomes from systematic reviews relevant to the various endpoints relevant to MetS. Have the systematic reviews been done using similar approaches in the different papers? This is important for understanding how comparable the outcomes may be. In addition what is the strength of evidence found in the systematic reviews and can these be supported by biological mechanisms? This is relevant for the review as associations between exposure and an effect can be very different, and the strength of that association and whether that association can be further supported is relevant to being able to translate the outcomes of the systematic review into public health risks.

Details of the conducted scoping review have been extended in the Methods section. Since we have used scoping review methodology in this paper, we are not able to provide a detailed analysis of the quality of the data or to quantify outcomes. The purpose of this scoping review was to present currently existing literature on the topic without deeper analysis of biological mechanisms etc.

  • Lines 169 to 173 provides an overview and risk outcomes based on the evidence presented in the section. This is the only chemical where overview/risk outcomes are presented. Could a similar overview discussion (few sentences) be included for all the other chemical sections - this would also support the discussion and conclusions.

Rather than adding a similar paragraph at the end of each chemical, we have moved this paragraph to the Discussion and added similar information from other chemicals there as well.

  • Line 313, do you mean PAH metabolites in urine samples?

Yes, we mean PAH metabolites. This has been clarified in the text.

  • Section 3.6.1 should be consistent with the use of the abbreviation "As" for arsenic. It looks a bit odd at the start of a sentence, but some sentences include both the abbreviation and the word, which also looks a bit strange.

We agree that abbreviation of arsenic ‘As’ can be misleading at the beginning of the sentence. We have edited text to write As as arsenic for clarity.

  • Is it possible to include a summary table that outlines the strength of the associations identified for each chemical and health outcome - perhaps that is what Figure 1 may have been trying to do, but it would be good to have the outcomes summarised further in the discussion.

Since this a scoping review, we are not able to draw conclusions about the strength of the evidence only provide an overview of known associations as presented in Figure 1. Discussion already had covered presented results from the perspective of MetS components. We have added a paragraph summarizing the outcome also from the chemical perspective in relation to MetS.

Reviewer 3 Report

Lines 11-12: Is Met just isolated to EUR, or is it prevalent in most countries now?

Lines 18-21: This sentence needs to be reworded into two sentences.

Why limit the review to just European populations? The authors are substantially limiting the significance of the review by excluding articles from other countries' populations.

Methods: details regarding the search criteria are severely lacking and need to be thoroughly explained. I understand that this review was not a systematic review. However, summarizing the review's scope allows readers better to understand the significance of the reviewed substances/topics.

Throughout the review, there are several instances where the cited articles do not match up to the scope of the paragraph/text. Ex: lines 149 – 153 reference (33) does not include Bisphenol A or the topic of "BPA associations with leptin and adiponectin suggest that adipokines may be more sensitive biomarkers of early metabolic impairment among children." Make sure and check references throughout.

Author Response

Thank you for taking time to review our manuscript and providing valuable feedback on it. Please, see below detailed responses to your comments/recommendations.

Lines 11-12: Is Met just isolated to EUR, or is it prevalent in most countries now?

MetS is a significant health burden in Europe as well as in most of the developed countries. Reason we are referring here to Europe, is that also selected chemicals evaluated in this paper are those prioritized within the European project HBM4EU.

Lines 18-21: This sentence needs to be reworded into two sentences.

Sentence has been edited and split into two sentences.

Why limit the review to just European populations? The authors are substantially limiting the significance of the review by excluding articles from other countries' populations.

Review is not limited to European populations. As it can be seen from the list of references, literature review included studies worldwide. We assume this comment refers to the sentence in the Abstract (line 22-23). With this sentence we mean that currently existing literature and studies are mainly outside Europe (see Discussion lines 558-563), and therefore, more standardized studies from Europe would be needed. We have edited the sentence in the abstract to clarify this.

Methods: details regarding the search criteria are severely lacking and need to be thoroughly explained. I understand that this review was not a systematic review. However, summarizing the review's scope allows readers better to understand the significance of the reviewed substances/topics.

Details of the conducted scoping review have been extended in the Methods section.

Throughout the review, there are several instances where the cited articles do not match up to the scope of the paragraph/text. Ex: lines 149 – 153 reference (33) does not include Bisphenol A or the topic of "BPA associations with leptin and adiponectin suggest that adipokines may be more sensitive biomarkers of early metabolic impairment among children." Make sure and check references throughout.

Thank you for pointing out this mistake. This mistake has been corrected and correct references has been changed to replace the wrong one. We have checked all the references in the text to ensure that right ones are referred to.